Gestational overweight decreased risk of antepartum hemorrhage in pregnant women with complete placenta previa: a retrospective study

Yang Jie 1
Ye Shaoxin 2
Xuan Bihua 1
Liu Zhengping 1 2
http://orcid.org/0000-0003-2773-9166 Fan Dazhi 1 2 fandazhigw@163.com
1 Department of Obstetrics, The Affiliated Foshan Women and Children Hospital, Guangdong Medical University , Foshan, Guangdong , China
2 Foshan Fetal Medicine Research Institute, The Affiliated Foshan Women and Children Hospital, Guangdong Medical University , Foshan, Guangdong , China
Oliveira Sonia
Electronic publication date: 2025 Feb 25
Publication date: 2025
Volume: 13
Electronic Location ID: e19091
Received 2024 Aug 13; Accepted 2025 Feb 11
Copyright: © 2025 Yang et al.
Copyright year: 2025
Copyright holder: Yang et al.
License: This is an open access article distributed under the terms of the Creative Commons Attribution License, which permits unrestricted use, distribution, reproduction and adaptation in any medium and for any purpose provided that it is properly attributed. For attribution, the original author(s), title, publication source (PeerJ) and either DOI or URL of the article must be cited.
License URL: https://creativecommons.org/licenses/by/4.0/

Keywords: Antepartum hemorrhage, Gestational overweight, Complete placenta previa, Risk, Pregnancy

Funding: Foshan 14th Five-Year Medical Key Specialty Construction Project (Nursing Care) Foshan Health Bureau Medical Scientific Research Project 20230814A010028, 20240720A010269 and 202532031004 Foshan Self-Funded Science and Technology Innovation Project 2220001004010 Special Project for Clinical and Basic Sci&Tech Innovation of Guangdong Medical University GDMULCJC2024137 This research was supported by the Foshan 14th Five-Year Medical Key Specialty Construction Project (Nursing Care), Foshan Health Bureau Medical Scientific Research Project (No. 20230814A010028, No. 20240720A010269, and No. 202532031004), Foshan Self-Funded Science and Technology Innovation Project (No. 2220001004010), and Special Project for Clinical and Basic Sci&Tech Innovation of Guangdong Medical University (GDMULCJC2024137). The funders had no role in study design, data collection and analysis, decision to publish, or preparation of the manuscript.

==============================
Background

Antepartum hemorrhage (APH) is associated with perinatal mortality and maternal morbidity. Previous studies have reported that obesity in pregnancy adversely influences both fetal and neonatal outcomes. This study aimed to investigate gestational overweight and the risk of APH in pregnant women with complete placenta previa (CPP).

Methods

This was a retrospective cohort study of pregnant women with CPP delivery at our hospital from 2013 to 2015. Outcomes were stratified according to APH and non-APH.

Results

Of 193 pregnancies with CPP, 40.4% (78) were diagnosed with APH. Maternal weight and BMI at delivery were significantly decreased in women with APH (61.15 ± 8.73 vs. 65.22 ± 7.80, 24.47 ± 3.12 vs. 26.21 ± 2.85; P = 0.001, P = 0.001; respectively), and the prevalence of overweight at delivery was higher in the non-APH group compared to those in the APH group (54.9% (62) vs. 39.7% (27); OR 2.18; 95%CI [1.16–4.11]). After adjusting for gestation week and other potential confounding factors, maternal weight and BMI were associated with the APH (OR 0.95, 95%CI [0.91–0.99]; 0.85, 95%CI [0.75–0.97], respectively).

Conclusion

Appropriate weight gain during pregnancy may decrease the risk of antepartum hemorrhage in pregnant women with complete placenta previa.

Introduction

Antepartum hemorrhage (APH) defined as bleeding from the genital tract in the second half of pregnancy, was associated with perinatal mortality and maternal morbidity (Dinho et al., 2024). A recent study showed that placenta previa was associated with a high risk of APH, constituting 53% of all such cases (Im et al., 2023). Women with placenta previa were at increased risk of maternal bleeding and emergency preterm cesarean section leading to perinatal complications (Fan et al., 2017b, 2017c; Kuribayashi et al., 2021; Mitsuzuka et al., 2022). Our previous study showed that placenta previa has been a high-burden disease in pregnant women (Fan et al., 2016, 2024b).

Complete placenta previa (CPP) was a serious complication of late pregnancy and it can unexpectedly lead to catastrophic blood loss during cesarean delivery and massive post-partum hemorrhage, defined as when the placenta covers the internal cervical os completely (Yue et al., 2024; Fan et al., 2019). Therefore, maternal mortality and morbidity were significantly increased due to complete placenta previa. In both high and low-resource setting, obesity is now a major health concern in pregnancy and has reached epidemic proportions globally (Zeng et al., 2023; Liu et al., 2024). Previous studies have reported that obesity in pregnancy influences both fetal and neonatal outcomes (Gunes et al., 2025; Cochrane et al., 2024). However, whether obesity or overweight during pregnancy has an effect on pregnancy outcomes, including antepartum hemorrhage, in women with complete placenta previa is not clear yet. In order to better illustrate the issue and fill the gap, we conducted this study.

The aim of this study was 1) to investigate gestational overweight and the risk of APH in pregnant women with complete placenta previa, and 2) to compare maternal and neonatal outcomes in women with complete placenta previa complicated with APH vs. those without APH episodes. The results will provide important reference for medical workers to formulate individualized weight gain goals to optimize maternal and fetal health for pregnant women with complete placenta previa. To the best of our knowledge, this is the first study to evaluate the results in pregnancies complicated with complete placenta previa in mainland Chinese population.

Materials and Methods

A retrospective analysis was performed through the review of the medical charts of women with complete placenta previa who had given birth at the Affiliated Foshan Women and Children Hospital, Guangdong Medical University from January 2013 to December 2015. Complete placenta previa was defined as complete coverage by the placenta of the internal os, and diagnosed by experienced obstetricians based on a serial transvaginal ultrasongraphic scans and confirmed at the time of delivery.

Women were divided into two groups that were compared: 1) women with APH; 2) women without APH, as a control group. APH was defined as vaginal bleeding before labor, requiring hospitalization. The data was retrospective and observational, and the study protocol was approved by the Institutional Review Board of the Affiliated Foshan Women and Children Hospital, Guangdong Medical University (FSFY-Med-2019044). The following maternal and neonatal data were retrospectively collected from electronic medical records, surgical records and anesthetic records from the hospital: maternal outcomes including maternal age, height, prepregnancy body weight and weight at delivery, gravidity, parity, gestational age at delivery, complications of pregnancy, location of the placenta (anterior or posterior), intraoperative blood loss, intraoperative blood transfusion, and length of hospital stay (days), and neonatal outcomes including birth weight, Apgar score (1 min, and 5 min) and umbilical artery pH. The primary outcomes included weight gain, body mass index (BMI) gain, overweight, and obese, and the secondary outcomes included maternal and neonatal outcomes.

Prepregnancy body weight was reported at first visit (8 weeks of gestation). Studies showed that there was no significant change in pregnancy weight at this stage compared with that before pregnancy (Wang et al., 2022; Lin et al., 2022; Huang et al., 2024). Body mass index was defined as maternal weight divided by the square of height in kg/m2 and categorized as normal weight for 18.5 to 24.9 kg/m2, overweight for 25.0 to 29.9 kg/m2, and obese for above 30.0 kg/m (Lin et al., 2019, 2023). Obstetrical complication including diabetes, hypertension, preeclampsia, anemia, and cardiovascular disease, based on diagnosis by attending medical doctor, was abstracted from obstetrical records. Gestational age was based on the last menstrual period and confirmed by sonographic examination during the first trimester. Placenta accreta was diagnosed using color Doppler ultrasound and confirmed by histological at the time of delivery (Liu et al., 2018; Feng et al., 2022).

Statistical analysis

Categorical data were reported as numbers and percentages (%), and descriptive data were expressed as means ± standard deviations (SD) or median (P25–P75), as appropriate. Comparisons between groups were performed using the t-test for normality and using the Wilcoxon rank sum test for non-normality. Statistical significance was calculated using the chi-square test for differences in qualitative variables. Fischer’s Exact Test was calculated when a single cell in a 2 * 2 contingency table had an expected frequency less than 5. For variables with a set of different categories, Mantel-Hanszel chi-square test for linear trend was used. Crude and adjusted odds ratios (OR), with 95% confidence intervals (95% CI), were also calculated by logistic-regression model in order to find independent association between variables of interest. The selection process for confounding variables in the logistic regression analysis are based on 1) the effect of variables on clinical outcomes, and 2) our previous and other similar studies (Fan et al., 2021b, 2021a, 2023; Post et al., 2023; Kayem et al., 2024). The SPSS Version 11.0 (SPSS Inc., Chicago, IL, USA) statistical package was used to analyze data. A probability value of <0.05 was considered to indicate statistical significance.

Results

There was a total of 31,844 deliveries during this period. One hundred and ninety-three (0.61%) women were diagnosed with complete placenta previa. Cesarean delivery was performed in all women. The rate of complete placenta previa was 0.58% (56/9,734), 0.56% (61/10,909) and 0.68% (76/11,201) in 2013, 2014 and 2015, respectively. However, no significant change was found in the rate over time (χ2trend = 0.982, P = 0.327).

A summary of maternal demographic and clinical characteristics was reported in Table 1. Of the 193 included participants, 78 (40.4%) were diagnosed with antepartum hemorrhage. Forty-three (22.3%) women were primiparas, eighteen (9.3%) women were prior cesarean delivery, ninety-seven (50.3%) women were prior abortion, and 153 women (79.3%) were accompanied by at least one pregnancy complications. Nonetheless, there were no significant differences in above maternal characteristics between women with APH and in those without APH.

Table 1 Maternal characteristics of the study groups.

	Total (n = 193 (100))	APH (n = 78 (40.4))	Non-APH (n = 115 (59.6))	P-values	
Age (years)	31.58 ± 5.57	31.41 ± 6.02	31.69 ± 5.27	0.725	
Maternal height (cm)	157.95 ± 4.63	158.01 ± 4.78	157.90 ± 4.54	0.871	
*Prepregnancy weight (kg)	54.10 ± 7.82	52.63 ± 8.41	54.85 ± 7.47	0.252	
*Prepregnancy BMI	21.44 ± 2.89	20.85 ± 3.02	21.74 ± 2.80	0.200	
*Prepregnancy Overweight (n)	8 (10.1)	2 (7.4)	6 (11.5)	0.709	
*Weight gain (kg)	10.30 ± 4.56	9.01 ± 4.03	10.95 ± 4.71	0.071	
*BMI gain (kg/m2)	4.09 ± 1.78	3.60 ± 1.65	4.36 ± 1.80	0.072	
Maternal weight (kg)	63.70 ± 8.38	61.15 ± 8.73	65.22 ± 7.80	0.001	
Maternal BMI (kg/m2)	25.56 ± 3.07	24.47 ± 3.12	26.21 ± 2.85	0.001	
Maternal BMI class					
Normal weight	78 (43.1)	38 (55.9)	40 (35.4)	Ref	
Overweight	89 (49.2)	27 (39.7)	62 (54.9)	0.017	
Obese	14 (7.7)	3 (4.4)	11 (9.7)	0.081	
Primiparous (%)	43 (22.3)	14 (18.0)	29 (25.2)	0.291	
Prior cesarean delivery (%)	18 (9.3)	9 (11.5)	9 (7.8)	0.452	
Prior abortion (%)	97 (50.3)	37 (47.4)	60 (52.2)	0.518	
#Complication (%)	153 (79.3)	64 (82.1)	89 (77.4)	0.474	
Length of hospital stay (days)	8.00 (6.00–12.50)	10.00 (5.75–18.00)	7.00 (6.00–10.00)	0.014	
Notes:

BMI: body mass index.

Data is presented as mean ± standard deviation, median (range), percentage (number).

* Only eighty participants provided data.

# At least one pregnancy complication, including diabetes, hypertension, preeclampsia, anemia, and cardiovascular disease.

No significant differences were evident in maternal age and maternal height between women with APH and without APH. However, maternal weight and BMI at delivery were significantly decreased in women with APH (61.15 ± 8.73 vs. 65.22 ± 7.80, 24.47 ± 3.12 vs. 26.21 ± 2.85; P = 0.001, P = 0.001, respectively), and the prevalence of overweight at delivery was higher in the non-APH group compared to those in the APH group (54.9% (62) vs. 39.7% (27); OR 2.18; 95%CI [1.16–4.11]), although there were no significant differences between the two groups when the prepregnancy weight, BMI, overweight, and weight, BMI gain, and maternal obese were compared. Length of hospital stay was also greater in women with APH (10.00 (5.75–18.00) vs. 7.00 (6.00–10.00); P = 0.014).

Perinatal outcomes in women with APH and without APH were shown in Table 2. Sixty women (31.1%) were anterior placenta position, and sixteen women (8.3%) were placenta accreta. Intraoperative blood loss was 530 (400–800) ml, and one hundred and five women (54.5%) needed for intraoperative blood transfusion. There were no significant differences in above characteristics between women with APH and without APH, except in the blood transfusion, which was higher in women with APH (53 (67.9%)) than in those without APH (52 (45.2%)) (OR 2.57; 95%CI [1.41–4.68]).

Table 2 Perinatal outcomes of the study groups.

	Total (n = 193 (100))	APH (n = 78 (40.4))	Non-APH (n = 115 (59.6))	P-values	
Anterior placenta (%)	60 (31.1)	26 (33.3)	34 (29.6)	0.636	
Placenta accreta (%)	16 (8.3)	7 (9.0)	9 (7.8)	0.795	
Blood loss (ml)	530 (400–800)	572.50 (400.00–853.75)	500.00 (400.00–745.00)	0.238	
Need for blood transfusion (%)	105 (54.4)	53 (67.9)	52 (45.2)	0.002	
Gestational age (weeks)	35.95 ± 2.27	34.54 ± 2.41	36.91 ± 1.56	0.001	
No. before 37 weeks	124 (64.2)	71 (91.0)	53 (46.1)	0.001	
No. before 34 weeks	32 (16.6)	28 (35.9)	4 (3.5)	0.001	
Weight (g)	2,673.92 ± 535.06	2,370.36 ± 533.34	2,879.82 ± 429.20	0.001	
pH	7.28 ± 0.08	7.28 ± 0.08	7.29 ± 0.08	0.551	
Apgar < 7/1 min	21 (10.9)	12 (15.4)	9 (7.8)	0.106	
Apgar < 7/5 min	3 (1.6)	1 (1.3)	2 (1.7)	0.999	
Note:

Data is presented as mean ± standard deviation, median (range), percentage (number).

In neonatal outcomes, gestational age in women with APH was lower than in those without APH (34.54 ± 2.41 vs. 36.91 ± 1.56; P = 0.001). Consequently, the incidence of preterm delivery was higher in women with APH than in those without APH (91.0% (71) vs. 46.1% (53); OR 11.87; 95%CI [5.03–28.00]), with a higher incidence of delivery before 34 weeks of gestation in women with APH (35.9% (28) vs. 3.5% (4); OR 15.54; 95%CI [5.17–46.66]). The weight of birth was lighter in women with APH (2,370.36 ± 533.34 vs. 2,879.82 ± 429.20; P = 0.001). However, no significant was found in the incidence of Apgar scores <7 at 1 and 5 min and umbilical arterial pH between women with APH and without APH.

Logistic-regression analyses were performed to assess the association between APH and potential variables. Results also showed that maternal weight, maternal BMI, need for blood transfusion, length of hospital stay, and infant birth weight were associated with APH. However, after adjusting for gestation week and other potential confounding factors, only maternal weight and BMI were associated with the APH (OR 0.95, 95%CI [0.91–0.99]; 0.85, 95%CI [0.75–0.97], respectively) (Table 3, Fig. 1).

Table 3 Logistic regression model to assess the APH and potential variables.

Covariates	Crude	Adjusted*	
OR	95% CI	P-values	OR	95% CI	P-values	
Weight (kg)	0.939	[0.903–0.977]	0.002	0.952	[0.909–0.996]	0.033	
BMI (kg/m2)	0.814	[0.727–0.912]	0.001	0.847	[0.744–0.965]	0.012	
Overweight (n)	0.433	[0.234–0.800]	0.009	0.532	[0.254–1.115]	0.094	
Need for blood transfusion (%)	2.568	[1.408–4.684]	0.002	1.458	[0.708–3.002]	0.306	
Length of hospital stay (days)	1.068	[1.023–1.114]	0.003	1.036	[0.993–1.080]	0.102	
Weight (g)	0.998	[0.997–0.998]	0.001	0.999	[0.998–1.000]	0.056	
Note:

* Adjust for gestation week, primiparous, prior cesarean delivery, prior abortion, and complication.

Figure 1 Comparison of maternal weight and BMI between APH and non-APH groups.

Discussion

Our results based on almost two hundred women with complete placenta previa demonstrated that gestational overweight may reduce the risk of antepartum hemorrhage in women with complete placenta previa. We also further confirmed that the incidence of preterm birth was higher in antepartum hemorrhage group compared to those without group in women with complete placenta previa.

The incidence of antepartum hemorrhage appears to have increased in relationship to the increasing rate of endometrial damage in pregnant women, and risk factors for endometrial damage include increasing maternal parity, induced labor, artificial abortion and the number of previous cesarean deliveries (Cecchino & García-Velasco, 2018; Long et al., 2021). However, no significant difference was observed in prior cesarean delivery between the APH and without APH groups in this study. This may be because of the relatively low prior cesarean delivery of our subjects, only eighteen (9.3%) women were prior cesarean delivery.

Although the mechanism of APH was uncertain, it appeared to be attributable to separation of the placenta from the underlying decidua resulting from cervical dilation, cervical effacement, contractions, and advancing gestational age (Fan et al., 2024a; Wei & Cheng, 2024). It was reported that APH was significantly more prevalent in women with complete placenta previa (He et al., 2023). In order to reduce the perinatal complications, the Royal College of Obstetricians and Gynaecologists recommended that women with complete placenta previa with previous bleeding events should be admitted at or after 34 weeks’ gestation, and that outpatient care should be considered for those without previous antepartum bleeding episodes (Jauniaux et al., 2019a). The recommend will decrease the risk of gestational complications (including APH) in pregnant women with complete placenta previa. However, the incidence of preterm delivery is higher in women with APH than in those without APH, and our results were also further proved this phenomenon.

Preterm birth, which is birth before 37 completed weeks of pregnancy, is a leading cause of a baby does not gain the appropriate weight before birth and a leading cause of infant mortality and neonatal morbidity (Zhang et al., 2019; Liang et al., 2024). The earlier the baby is born, the lower the birth weight, and the more increased morbidity, mortality, and cost in infants (Ahmed et al., 2024). Although still controversial, many reports revealed that cervical length in the third trimester was associated with APH in patients with complete placenta previa (Huang et al., 2022; Hessami et al., 2024).

In this study, we found that overweight in pregnancy decreased the risk of APH and reduced the risk of preterm birth in women with complete placenta previa. Previous studies have also got similar results in women without complete placenta previa (Riley et al., 2016; Girsen et al., 2016). Riley et al. (2016) showed that obesity was associated with reduced risk of spontaneous preterm birth among multiparous women. In addition, Girsen et al. (2016) demonstrated that the risk for preterm birth increased with the severity of underweight, and that the relation persisted even after adjusting for maternal characteristics. The explanation for this observation remains uncertain; however, it may be that overweight women with these conditions have a stronger association with underlying mechanisms wherein belly fat can decrease uterine sensitivity during the third trimester of pregnancy. In biological terms, a plausible explanation is that the reduced risk of preterm birth may be due to an accelerated expansion of T-regulatory (T-reg) cells during the second pregnancy (Moore et al., 2022). This increase in T-reg cells could, in turn, contribute to a lower risk of preterm birth.

It should be noted that the proportion of PAS is relatively high, reaching 8.3% in this study, much higher than 0.22% in mainland China (Fan et al., 2017a) and 0.17% in the worldwide (Jauniaux et al., 2019b). The two main risk factors for PAS are prior cesarean section and placenta previa. Meanwhile, advancing maternal age, in vitro fertilization, multiparity, smoking and a short interval between a previous cesarean delivery and the subsequent pregnancy can also increase the risk of PAS (Wang et al., 2018; Liu et al., 2021; Premkumar et al., 2025). The high incidence of placenta accreta spectrum disorders in a dataset where only 18 (9.3%) of cases had previous cesarean deliveries further underscores the significant role that placenta previa plays in the development of PAS.

The findings from this research will serve as a crucial reference for healthcare providers in establishing individualized weight gain guidelines tailored to optimize maternal and fetal health for pregnant women diagnosed with complete placenta previa. By understanding the specific needs and risks associated with complete placenta previa, medical professionals can provide more targeted and effective prenatal care. This approach not only aims to improve immediate pregnancy outcomes but also supports long-term health benefits for both mother and child, emphasizing the importance of personalized medicine in obstetric practice. Additionally, these insights can inform the development of clinical protocols and patient education materials, ultimately contributing to better health management and decision-making during pregnancy.

The strength of this study was that it included a larger population of nearly two hundred compared to other prior studies, although it was still a small size because of the rarity of the disease. Given the complexity of the influencing factors, there were some limitations in our study. Firstly, as a tertiary specialized hospital, pregnant women with complete placenta previa often referred to our hospital, which will overestimate the incidence of complete placenta previa and affect the representativness and generalizability of the results. Secondly, only eighty women provide prepregnancy weight in this study. Although the studies showed no significant changes at both time points (Lin et al., 2022, 2023; Huang et al., 2024), we chose to use the maternal weight at the first prenatal visit (8 weeks of gestation) rather than the before pregnancy weight. Thirdly, data inconsistencies between the old and new hospital systems limited the study period. To address this, we are manually reorganizing data from the old system into a format compatible with the new system to facilitate future research. Lastly, the main limitation of this study was a retrospective assessment, which inherently restricts the ability to establish causality and may be subject to biases such as recall bias or selection bias. Future multicenter studies, including ourselves, should aim to address the aforementioned limitations by focusing on preconception weight management, distance that the placenta overlies the internal os, and investigating the underlying biological mechanisms. These studies should also incorporate extended periods of data collection to provide more robust and comprehensive insights.

Conclusions

Overall, our current data suggest that appropriate weight gain during pregnancy may be associated with a reduced risk of antepartum hemorrhage in women with complete placenta previa. The findings will serve as a crucial reference for healthcare providers in establishing personalized weight gain targets, thereby optimizing maternal and fetal health outcomes for pregnant women diagnosed with complete placenta previa.

Supplemental Information

Supplemental Information 1 Raw data.

Supplemental Information 2 Supplemental tables.

Supplemental Information 3 New author Shaoxin Ye’s contributions.

Additional Information and Declarations

Competing Interests

The authors declare that they have no competing interests.

Author Contributions

Jie Yang performed the experiments, prepared figures and/or tables, authored or reviewed drafts of the article, and approved the final draft.

Shaoxin Ye performed the experiments, prepared figures and/or tables, authored or reviewed drafts of the article, and approved the final draft.

Bihua Xuan performed the experiments, analyzed the data, authored or reviewed drafts of the article, and approved the final draft.

Zhengping Liu conceived and designed the experiments, authored or reviewed drafts of the article, and approved the final draft.

Dazhi Fan conceived and designed the experiments, analyzed the data, prepared figures and/or tables, authored or reviewed drafts of the article, and approved the final draft.

Ethics

The following information was supplied relating to ethical approvals (i.e., approving body and any reference numbers):

This study has been performed in accordance with the Declaration of Helsinki. This study was approved by the Ethic Committee of the Affiliated Foshan Women and Children Hospital, Guangdong Medical University (FSFY-Med-2019044). All cases were routinely and retrospectively collected and datasets were fully anonymized prior to analysis, and the data collection was registered with the audit department. The Institutional Review Board of the Affiliated Foshan Women and Children Hospital, Guangdong Medical University waived the need for consent, informed consent, written or verbal, from all participants approval (FSFY-Med-2019044).

Data Availability

The following information was supplied regarding data availability:

Raw data are available in the Supplemental Files.

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
