# Peer review of "Gestational overweight decreased risk of antepartum hemorrhage in pregnant women with complete placenta previa: a retrospective study"

_PeerJ, doi:10.7717/peerj.19091_

## Round 0.1 · original submission · Major Revisions

Dear authors, thank you for your submission. Your work is interesting but requires some revisions before approval for publication. Please, refer to the reviewers' comments for further details. Additionally, I recommend reviewing more recent literature, as many if not all of the references you’ve included appear to be "outdated" (from over 4-5years ago).

Reviewer 1 ·

Basic reporting

Minor correction is needed.
Line 18, "in the world" should be deleted.
Line 63 & 68, hospital name be described instead of 'our hospital'.
Line 165, noun after spontaneous is missing. Please add it.

Experimental design

No comment

Validity of the findings

No comment

Additional comments

APH in Complete placenta previa important for maternal and child health. The author find out risk assessment of APH by gestational overweight.

Reviewer 2 ·

Basic reporting

The study addresses a clinically significant topic and contributes valuable data on gestational overweight and its association with antepartum hemorrhage (APH) in women with complete placenta previa (CPP). The manuscript is well-structured, and the key findings are clearly presented. It is professionally written and mostly uses clear, unambiguous language. Figures and tables are relevant and well-labeled, and the raw data provided supports transparency.

However, the writing can benefit from minor improvements in language and grammar to enhance clarity and readability. For example, phrases like "The incidence of preterm delivery does be higher" should be revised to "The incidence of preterm delivery is higher." Editing for linguistic precision would improve the manuscript’s readability for an international audience.

The introduction effectively sets the context, but it could better emphasize how this study fills existing gaps in the literature to strengthen its impact. Expanding on the justification for the study would highlight its novelty.

Additionally, consistency in the reporting of statistical results (e.g., formatting p-values uniformly) would enhance the manuscript’s technical presentation.

While the conclusions are well-supported by the results, further exploration of the biological mechanisms and broader implications would add depth to the discussion.

Experimental design

The strength of this study that is original, addressing a relevant gap in research, with very excellent detailed methodology that allows for reproducibility, and ethical considerations are appropriately addressed.


The weakness of this study is that retrospective design use which limits causal interpretations. Also, the low number of participants (e,g 193) providing pre-pregnancy weight data could introduce bias.Howevere, the authors address the limitation regarding the retrospective design and the sample size in the discussion section. They acknowledge that the retrospective nature of the study limits causal interpretations and note the relatively small sample size as a constraint, particularly in relation to the number of participants providing pre-pregnancy weight data. They also highlight that the study's sample may not fully represent the general population due to the tertiary care setting.
The important weakness of this stdy is that The authors do not explicitly address why the study period was limited to deliveries from January 2013 to December 2015 instead of including more recent years, why????

My sugegetion:
1- Discuss the potential impact of referral bias due to the tertiary center setting on the generalizability of results.
2- Justify the decision to use maternal weight on admission instead of pre-pregnancy weight, as this is a limitation.

Validity of the findings

Strengths: Statistical analyses are comprehensive, and the results are presented with adequate detail. The association between maternal weight/BMI and APH is well-supported.
Weaknesses: Some analyses, such as the impact of gestational overweight on neonatal outcomes, could be explored further.
Suggestions:
1/Add a sensitivity analysis or stratified analysis to evaluate potential confounders, such as the influence of gestational diabetes or hypertensive disorders.
2/ Provide more discussion on the clinical relevance of the findings, especially for obstetric management.

Additional comments

The introduction effectively reviews relevant literature. However, it should better emphasize how the study's findings can influence clinical practice or guidelines. Please make your introuduction more informative about the topic.

The results are clearly organized, but further exploration of the relationship between maternal BMI categories and specific outcomes (e.g., gestational age or blood transfusion rates) could be beneficial.

The discussion integrates findings well with existing literature. Including potential biological mechanisms for the protective effect of overweight on APH would enrich the narrative. The limitations section is appropriately addressed, but suggesting strategies to mitigate these limitations in future research would be helpful.

The conclusion is concise but should provide a stronger take-home message about the implications of gestational overweight for clinical practice.

·

Basic reporting

General remarks:
- Please improve standard of English language
- Consider using latest terminology for placenta previa.
Placenta previa is a placenta overlying the internal os for one or more millimeters.
Otherwise low lying placenta

Experimental design

See below

Validity of the findings

see below

Additional comments

Introduction:
Title suggest primary outcome/aim of the study to be to describe the association between maternal BMI and APH. However, the first aim described in introduction: comparison of maternal and neonatal outcomes.

Material and methods:
Please describe primary and secondary outcomes (and adjust title and introduction accordingly)

Please provide insight into the selection process for confounding variables in the logistic regression analysis.

Results
- Line 100: What is meant by a “clinically significant” complete placenta previa.
- Focus on clinically relevant outcomes - for instance, consider leaving out: incidence of male [fetus]
- Please provide guidance for clinical practice since the mean BMI difference is small (1.7 BMI points), APH compared to non-APH.

Table 1:
- If possible provide information on distance that the placenta overlies the internal os.
(And consider adding it as a possible confounding variable in the regression analysis.)

Table 2:
- Placental characteristics description/analysis do not fit aim of study. Not added in the regression analysis, consider leaving out.

- Please provide insight into the high proportion of PAS (especially when taking into account only 18 women had a prior cesarean section).

Table 3:
- Please provide extra information on the variables in the logistic regression analysis.
o Is BMI added into the model as a continuous variable?
(Which would mean that every single BMI point increase is associated with a 15% odds decrease?)
- What is the reference group for “Overweight”?
- What is the added value of analyzing weight, BMI and classification of overweight separately?

Discussion:
Line 143-145: Please provide a reference

Conclusion:
Lines 186-187: “…gestational overweight may decrease the risk of antepartum hemorrhage…”
Gestational overweight was not significant in de multivariate model. Consider changing conclusion.

---

## Round 0.2 · Minor Revisions

Dear authors,

Thank you for your revisions. There still remains one area of concern: the study period limitation (2013–2015). The authors explained that data inconsistency between old and new hospital systems constrained the study period, which is a reasonable justification. Nevertheless, you did not specify whether this issue has since been resolved or if future research will incorporate more recent data. While some rationale has been provided, it is advisable to have this situation perfectly clarified (and such be also clear in the manuscript itself).

Reviewer 2 ·

Basic reporting

Well addressed.

Experimental design

Everything was well addressed except for the study period limitation (2013–2015). The authors clarified that data inconsistency between old and new hospital systems limited the study period, and while their explanation is reasonable, they could have briefly stated whether this issue has since been resolved or if future research will address newer years. Although they provided some justification, I am still not fully persuaded.

Validity of the findings

Well addressed.

Additional comments

Well addressed and provides valuable clinical implications.

---

## Round 0.3 · accepted · Accept

Dear authors,

i am not accepting your work for publication in PeerJ. Thank you for your hard word and congratulations.